# Direct Oral Anticoagulant Drugs: On the Treatment of Cancer-Related Venous Thromboembolism and their Potential Anti-Neoplastic Effect

**DOI:** 10.3390/cancers11010046

**Published:** 2019-01-05

**Authors:** Francesco Grandoni, Lorenzo Alberio

**Affiliations:** 1Division of Haematology and Haematology Central Laboratory, CHUV, University Hospital of Lausanne, 1011 Lausanne, Switzerland; francesco.grandoni@chuv.ch; 2Faculty of Biology and Medicine, University of Lausanne, 1011 Lausanne, Switzerland

**Keywords:** cancer, venous thromboembolic event, CAT, direct oral anticoagulant drugs, DOACs, protease-activated receptors, PARs

## Abstract

Cancer patients develop a hypercoagulable state with a four- to seven-fold higher thromboembolic risk compared to non-cancer patients. Thromboembolic events can precede the diagnosis of cancer, but they more often occur at diagnosis or during treatment. After malignancy itself, they represent the second cause of death. Low molecular weight heparins are the backbone of the treatment of cancer-associated thromboembolism. This treatment paradigm is possibly changing, as direct oral anticoagulants (DOACs) may prove to be an alternative therapeutic option. The currently available DOACs were approved during the first and second decades of the 21st century for various clinical indications. Three molecules (apixaban, edoxaban and rivaroxaban) are targeting the activated factor X and one (dabigatran) is directed against the activated factor II, thrombin. The major trials analyzed the effect of these agents in the general population, with only a small proportion of cancer patients. Two published and several ongoing studies are specifically investigating the use of DOACs in cancer-associated thromboembolism. This article will review the current available literature on the use of DOACs in cancer patients. Furthermore, we will discuss published data suggesting potential anti-cancer actions exerted by non-anticoagulant effects of DOACs. As soon as more prospective data becomes available, DOACs are likely to be considered as a potential new therapeutic option in the armamentarium for patients suffering of cancer-associated thromboembolism.

## 1. Introduction

Thromboembolism represents a common complication of cancer. The risk of developing a venous thromboembolic event (VTE) associated with malignancy is estimated to be four to seven-fold higher compared to the normal population [1,2]. Between 20% and 30% of first thromboembolic events (TEs) are associated with malignancy [3]. Due to increasing aging of the general population, a much more rapid diagnosis of malignancies and a prolonged survival of patients, the incidence of thromboembolic events among cancer patients is increasing.

A VTE is the second cause of death in addition to the malignant disease itself [4]. Cancer patients who develop a TE have a shorter survival compared to those without [5,6,7]. Moreover, treatments of the underlying neoplastic disease can be delayed, hospitalisation stay prolonged and, eventually, health-care costs are raised. All these characteristics highlight the importance of a well-managed treatment. Such a treatment should be simple (improving patient’s compliance), cheap, and safe—with the smallest possible rate of recurrences and/or bleedings.

Low-molecular-weight heparins (LMWHs) are the treatment of choice for TEs in cancer patients. Nevertheless, their utilization in daily practice is far from being ideal. There is evidence that, before the advent of direct oral anticoagulants (DOACs), the adherence to the current guidelines, with LMWHs being the gold standard, is about 30–60% [8]. The mode of administration—subcutaneous injection—negatively influences compliance putting patients at risk of TE recurrence. The long-term use of LMWHs is associated with significant costs. Consequently, according to some data about 25% of cancer patients are already treated with DOACs and 30% are under vitamin K antagonists (VKAs) [9].

## 2. Treatment of Cancer-Associated Venous Thromboembolism: Current Guidelines

Many guidelines for the management of the cancer-associated thromboembolism (CAT) are published and regularly updated. The most recent updates date from 2016 by the American College of Chest Physicians (ACCP) [10], 2015 by the American Society of Clinical Oncology (ASCO) [11], 2015 by the Haemostasis and Thrombosis Task Force of The British Committee for Standards in Haematology [12], 2016 by the International Initiative on Thrombosis and Cancer (ITAC-CME) [13] and 2011 by the European Society of Medical Oncology (ESMO) [14]. Based on these guidelines, LMWHs are still the treatment of choice for TEs in patients with a malignancy. LMWHs are the preferred agent for both the initial and for the long-term treatment, which should last at least three months.

Currently, DOACs are not considered for the acute and long-term treatment of CAT in most guidelines. However, several guidelines recognize the potential of these drugs. For instance, the 2016 updated consensus recommendations from the International Initiative on Thrombosis and Cancer (ITAC-CME) consider DOACs for patients with stable disease not receiving systemic anticancer therapy [13]. The 2016 published ACCP guidelines state that there is no preference between DOACs and VKAs when LMWHs are not used. Since 2016, even if based on weak clinical evidence, DOACs are considered as a good alternative to LMWHs, particularly in the presence of problems (e.g. costs, painful injections) which could lead to poor long-term compliance. The recently published update 2018 of the clinical guidelines by the National Comprehensive Cancer Network (NCCN) consider edoxaban for the treatment of patients diagnosed with CAT with a level 1 of evidence [15]. Moreover, recommendations edited in June 2018 by the Scientific and Standardization Committee (SSC) from the International Society on Thrombosis and Haemostasis (ISTH) suggest the use of specific DOACs for cancer patients with an acute diagnosis of VTE. The recommendation considers the use of edoxaban or rivaroxaban as these two molecules were compared with LMWHs in randomized clinical trials. DOACs should only be administered in patients presenting with a low risk of bleeding, without any drug–drug interactions with systemic cancer treatment and after an in-depth discussion about the risks and benefits with the patients [16].

## 3. Low-Molecular-Weight Heparins (LMWHs)

LMWHs were developed in the late seventies and early eighties of the last century. They are obtained from the chemical or enzymatic depolymerization of heparin so that the molecules reach a weight of 4000–5000 Da. The anticoagulant effect is due to the activation of antithrombin, which promotes the inactivation of factor Xa (and to a lesser extent of thrombin). The major advantage of LMWHs was represented by the subcutaneous administration compared to intravenous administration of unfractionated heparin (UFH) with the same antithrombotic efficacy [17]. Their introduction in daily clinical practice started in the second part of the nineties after the publication of four randomized clinical trials between 1996 and 1997 confirming safety and efficacy to treat TEs compared to UFH [18,19,20,21].

UFH followed by VKAs were the standard of care before the introduction of LMWHs in patients with malignancy. Prandoni and colleagues showed an approximately double bleeding risk and a threefold thrombotic recurrence risk when patients with malignancy were treated with VKAs compared to non-cancer patients [22].

Five randomized trials compared VKAs to LMWHs for the treatment of cancer associated VTE: the CLOT trial [23], the LITE trial [24], the CANTHANOX trial [25], the ONCENOX trial [26] and the CATCH trial [27]. The first randomized clinical study (CANTHANOX trial) included 146 patients comparing warfarin versus enoxaparin. It showed that more patients treated with warfarin (21.1%) versus patients treated with LMWHs (10.5%) developed the complications defined as the primary combined endpoints at three months, namely recurrent VTE (4% versus 2.8%) or major bleeding (16% versus 7%). Of note, in this trial there were more bleedings than recurrent VTE. Due to slow recruitment, the study was interrupted early. The consequent lack of power of the study did not reveal a significant difference in the two groups [25].

The most important study which defined LMWHs as the standard of care was the randomized CLOT trial that included 672 patients with cancer being diagnosed six months prior to inclusion (squamous and basal cell carcinoma excluded) comparing dalteparin versus warfarin. LMWHs were superior to warfarin permitting the reduction of the risk of recurrent VTE (9% versus 17% at six months) without an increasing in bleeding events or in mortality [23].

The other LMWHs enoxaparin (ONCENOX trials including 102 patients [26], and the CANTHANOX trials including 146 patients [25]) and tinzaparin (LITE trials including 200 patients [24], and CATCH trial including 900 patients [27]), were compared to warfarin, as well. Of these trials, only the results of LITE study were able to show a statistically significant reduction on recurrent VTE with similar rates of bleeding and mortality at 12 months. In fact, even if the much bigger CATCH study showed a reduction of all VTE recurrence at six months, the difference was not statistically significant. However, the results of the CATCH study showed a significant reduction of recurrent proximal DVT under LMWHs. Therefore, the CATCH study is seen as confirming current guidelines.

A recent meta-analysis of 15 randomized controlled trials published in 2018 by Hakoum and colleagues confirmed the efficacy of LMWHs as the treatment of choice for the initial treatment of venous thromboembolism in patients with cancer [28]. However, LMWHs are far from being the ideal treatment of CAT taking into consideration the peculiar features of these patients. LMWH treatment requires daily injections, the dose must be weight-adjusted and withhold in case of severe thrombocytopenia. Moreover, management of patients whose renal function is inferior to 30 mL/min is difficult and LMWH treatment can be complicated by heparin-induced thrombocytopenia (HIT).

## 4. General Aspects of DOACs and Concerns in Cancer Patients

The currently available DOACs were approved during the first and second decades of the 21st century. Three molecules (apixaban, edoxaban and rivaroxaban) are targeting the activated factor X and one (dabigatran) is directed against the activated factor II, thrombin. They are employed to prevent and to treat TEs (deep vein thrombosis (DVT) or acute pulmonary embolism (PE)) or to prevent stroke and a systemic embolization in cases of non-valvular atrial fibrillation. Hemodynamically unstable patients, pregnant women, patients with extensive DVT and patients with severe renal impairment should not be treated with these agents. The most important advantages are represented by the absence of regular monitoring, the oral administration and a lower bleeding risk.

Their efficacy and safety in treating and preventing VTE were proved by randomized non-inferiority designed trials that compared the newer molecule with the standard treatment (LMWHs followed by VKAs). We briefly resume the most important aspects of these trials as they provided the first basic information on the utilization of DOACs in the subgroup of cancer patients.

Rivaroxaban showed a non-inferiority profile compared to VKAs with a similar rate of relapse of VTE and of bleeding in a total of 8281 patients included in two trials (3449 patients in the EINSTEIN-DVT trial and 4832 in the EINSTEIN-PE trial) [29,30].

Apixaban was non-inferior compared to the standard of treatment in the rate of recurrence and in mortality, and was superior in the rates of bleeding in a total of 5395 patients enrolled in the AMPLIFY trial [31].

Edoxaban was able to demonstrate similar results than warfarin in a total of 4118 patients enrolled in the trial of the Hokusai-VTE investigators [32].

Dabigatran was non-inferior in the capacity of preventing recurrence of VTE compared to the standard of treatment and in mortality as well in bleeding events. This was demonstrated in the acute VTE trials (RECOVER I) which included 2539 patients [33], in the 2589 enrolled patients of the acute VTE trials (RECOVER II) [34], and in the pooled analysis of RE-COVER I and II [35].

It is important to consider that these drugs have some limitations and some concerns arise, especially in the population of cancer patients. The management of bleeding in such patients is still an issue, not only because antidotes are limited. Patients with cancer often experience a modification of thrombocyte count mainly, but not only, due to treatments. LMWHs are usually given at full dose when platelet counts are >50 × 10^9^/L. With the exception of the AMPLIFY trial, which specified a platelet count of >100 × 10^9^/L as inclusion criterion, no precise cutoff was defined in the other studies. A platelet count >50 × 10^9^/L is generally considered adequate for therapeutic anticoagulation. Evidence of utilization of DOACs with lower platelets count is lacking.

Drug interactions with DOACs are of cardinal importance and clinicians are to be aware of this problem. The metabolism of DOACs relies on P-glycoprotein transport and on CYP3A4 pathway. Drugs that modify these metabolisms should be avoided. Some examples of agents inducing these pathways are dexamethasone, vinblastine, doxorubicin and paclitaxel. Drugs inhibiting these pathways are cyclosporine, tamoxifen, and tyrosine kinase inhibitors [36,37].

Renal and hepatic function influence the choice of the anticoagulant treatment. With the exception of apixaban—AMPLIFY trial excluded patients whose creatinine clearance was <25 mL/min—a minimal creatinine clearance of 30 mL/min is recommended to consider a treatment with DOACs. The same is true for the treatment with LMWHs. Even if not precisely defined—values of ASAT/ALAT >2 times the upper limit of normal or a bilirubin >1.5 times the upper limit of normal were considered as exclusion criteria in the main trials—a significant modification of liver function represents a contra-indication to DOACs.

Oral intake can be a concern for cancer patients. Rivaroxaban has to be administered with food and to ensure an adequate bioavailability has to be administered to the stomach via PEG tube, when required [38]. Apixaban can be employed independently of food intake, as well as via PEG. Edoxaban and dabigatran are to be administered as an intact capsule, the first one because of limited data and the second because of an increased bioavailability when the capsule is removed. Both are not recommended for utilization via PEG.

Finally, patients affected by cancer often suffer from gastro-intestinal alterations (i.e., lack of appetite, nausea, vomiting, diarrhea, mucositis). Therefore, not only the simple swallowing of a pill can be problematic, but these patients are also at an increased risk of gastro-intestinal bleeding and at a higher risk of a modified absorption or clearance (e.g., diarrhea which results in a diminished bioavailability) of DOACs [37,39]. A recently published review by Barr and Epps on the managing of DOACs nicely resumes the correct strategies to minimize errors by prescribing these drugs [40].

## 5. Evidence of Utilization of DOACs in Cancer Patients

In comparison to the data on treatment of VTE with LMWHs in cancer patients, evidence on the use of DOACs in this patient’s group is limited [41,42].

### 5.1. Subgroup Analysis

The only available data until 2017 were obtained from subgroups analysis of cancer patients of the main phase III randomized controlled trials comparing DOACs with VKAs in acute VTE. Altogether, these data were retrieved from about 1500 patients enrolled in such studies as having a known malignancy at baseline, which represent 6% of the patients treated with DOACs [43,44]. Consequently, because of the small sample size and the exploratory nature of the subgroups analysis, these data did not permit any reliable conclusions but offered a primary evaluation of the efficacy and safety.

In the subgroup analysis, 655 patients were randomly assigned to receive rivaroxaban versus VKAs (EINSTEIN-DVT and EINSTEIN-PE trials) [29,30,45], 771 patients assigned to receive edoxaban versus VKAs (Hokusai-VTE) [32,46], 335 patients assigned to receive dabigatran versus VKAs [33,47,48], and 159 patients assigned to receive apixaban versus VKAs [31,49]. We focus our attention on the subgroup of patients who presented an active cancer at enrolment or whose cancer was diagnosed during the study, in an attempt to consider a patient population as homogeneous as possible, excluding those patients with a previous history of cancer.

#### 5.1.1. Rivaroxaban

In the pooled rivaroxaban subgroup analysis [45], a total of 655 patients (8% of the 8281 included in EINSTEIN-DVT and EINSTEIN-PE trials) presented any active cancer. Of these, 462 (6%) had an active cancer at study inclusion and 193 (2%) were diagnosed during the study. Patients with a history of cancer (defined as not meeting the definition characteristic of these first two subgroups) were 469 (6%) and 7157 patients (86%) were never diagnosed with cancer.

Diagnosis of cancer or treatment within six months before inclusion, recurrent or metastatic cancer, including basal-cell or squamous-cell carcinoma of the skin, were the criteria defining active cancer at study inclusion. A new diagnosis of cancer after the randomization or a recurrence during the study defined patients with active cancer during the study. In the first subgroup (active cancer at study inclusion), rates of VTE recurrence was 2% in patients on rivaroxaban and 4% in patients on VKAs. Major bleeding occurred in the same proportion as VTE recurrence (2% under rivaroxaban and 4% on VKAs). In the second subgroup (cancer diagnosed during the study), rate of VTE recurrence were 10% in patients treated with rivaroxaban and 12% in patients treated with VKAs. Major bleeding occurred in 3% on rivaroxaban and in 7% on VKAs. The data suggested that rivaroxaban could be considered as an alternative of VKAs in patients with CAT. In all three subgroups, in fact, the conclusions for efficacy and safety did not significantly differ in patients receiving rivaroxaban or VKAs.

#### 5.1.2. Edoxaban

In the edoxaban subgroup analysis [46], 771 patients (9.4%) of the 8240 patients in the modified intention-to-treat and safety analysis of the Hokusai-VTE trial were defined as having had a history of cancer at study inclusion. Unlike rivaroxaban, patients were treated with an initial parenteral anticoagulation either with unfractionated heparin (UFH) or LMWHs for ≥5 days.

Patients were defined as having active cancer based on the clinical judgement of the local investigator. Furthermore, a review of all patients with a history of cancer were performed by an independent physician without knowledge of treatment group and patients. The data was collected in a post-hoc analysis. In this case, the definition of cancer was more accurate and defined precisely as either a presence of solid measurable cancer except non-melanoma skin cancer or the presence of a not in remission haematological malignancy. In the analysis of the patients with any history of cancer (*n* = 771), rates of VTE recurrence were 4% in patients on edoxaban and 7% in patients on VKAs. Rates of major bleeding were similar (3% of patients who received edoxaban and 3% of patients on warfarin). Clinically relevant bleeding (major and non-major) occurred in 12% in the edoxaban group and in 19% in the warfarin group.

In the pre-specified analysis of patients with active cancer (*n* = 208), rate of VTE recurrence were the same as in the groups with any history of cancer (4% for edoxaban and 7% for VKAs). Clinically relevant bleeding occurred at a higher rate in both arms, 18% for edoxaban and 25% for VKAs, respectively. In the group of patients diagnosed with cancer during the study (*n* = 175), rate of VTE recurrence were 17% (edoxaban) and 20% (VKAs). Similarly to rivaroxaban, these data suggested that edoxaban could be considered as an alternative of VKAs in patients with CAT as well.

#### 5.1.3. Dabigatran

In the dabigatran subgroup analysis 335 patients were included. This corresponds to 6.6% of the 5107 patients of the entire study population (RECOVER I/II trials) [48]. Active cancer at inclusion (*n* = 221) was defined as diagnosed within 5 years, except basal- or squamous-cell carcinoma of the skin, with any treatment within this period or any recurrent/metastatic disease. The remaining participants (*n* = 114) were patients with a diagnosis of cancer made during the study. Like edoxaban, an initial parenteral treatment with UFH, LMWHs or fondaparinux were conducted for ≥5 days.

In the subgroup of patients with active cancer, rate of VTE recurrence was 3.5% for dabigatran and 4.7% for VKAs. Major bleeding occurred in 3.8% on dabigatran and in 3.0% on VKAs. In the subgroup of patients diagnosed during the study, the rate of VTE recurrence was 8.5% for dabigatran and 13% for VKAs. The risk of major bleeding in this subgroup was 3.7% for dabigatran and 7.7% for VKAs. In the latter group, the rate of recurrence was higher than in the group of patients with active cancer at inclusion (hazard ratio (HR) 2.6, 95% confidence interval (CI) 1.1–6.2). In terms of efficacy and safety, there was no difference in patients receiving dabigatran or VKAs.

#### 5.1.4. Apixaban

Of the 5395 patients included in the AMPLIFY trial [31], 169 patients (3.1%) had an active cancer, whereas 365 patients (6.8%) had a history of cancer in the subgroup analysis [49]. Active cancer was defined as diagnosed or treated within the previous 6 months of inclusion. Patients with a history of cancer were defined as those with a diagnosis >6 months previous enrollment without receiving any treatment. It is to be noted that 25 patients diagnosed with cancer during the study were included in the cohort of the remaining 4861 patients (90.1%) without any active cancer nor a history of cancer. These cancers were identified based on the analysis of adverse event reports.

In the subgroup of patients with active cancer at study inclusion, the primary efficacy outcome (recurrent symptomatic VTE or VTE-related death) occurred in 3.7% for apixaban and 6.4% for VKAs. Major bleeding occurred in this subgroup in 2.3% under apixaban and 5.0% under VKAs. These findings suggest that apixaban is effective as VKAs in preventing VTE-recurrence without any significative difference in the safety profile.

### 5.2. Meta-Analysis

The data of at least six (semi-)systematic reviews and meta-analysis and of all the phase 3 data published between 2014 and 2015 illustrate that DOACs present a similar efficacy and safety profile as VKAs [44,50,51,52,53,54]. Altogether, the findings of DOACs issued from the four subgroups analysis and from their meta-analysis, showed at least a similar trend in the capacity of preventing recurrent VTE with a similar safety profile as VKAs in the treatment of CAT [43,44,54].

However, the subgroup analysis had many important limitations. First, the comparators of DOACs were VKAs, which are less effective in preventing recurrent VTE than LMWHs. Second, in the phase 3 clinical trials only 6% of patients were suffering from active cancer [44]. Third, there is also a paucity of data of the type of cancer, the staging of disease and the oncologic treatment in the different subgroup analysis. Forth, the state of active cancer at baseline was not well defined and differed from the trials evaluating LMWHs for CAT [54]. Therefore, patients with aggressive cancer (with end-organ dysfunction or reduced life expectancy) were excluded limiting the representability of patients at higher risk for cancer-associated thrombosis (roughly, 30% were under chemotherapy and only 15% to 30% were metastatic). For these reasons, based on the data coming from highly selected cancer population (subgroup analysis and their meta-analysis), it is not possible to draw firm conclusions on the safe utilisation of DOACs in cancer patients.

### 5.3. Observational Studies

Several observational studies describe the use of DOACs in cancer patients [55,56,57,58,59,60] and are summarized in a recently published systematic review [61]. The most frequently used treatments in these studies were enoxaparin and rivaroxaban. The duration of the utilization of DOACs was longer than that of LMWHs, possibly due to patient preferences for oral drugs or to lower costs of DOACs. With the exception of one study [55], the review reports lower rates of recurrent VTE in patients treated with DOACs in comparison to patients treated with LMWHs. In terms of safety (major bleedings and clinically relevant non-major bleedings, CRNMB), the results of the observational studies were heterogeneous. The main limitations of the observational studies are indications and patient selections bias by clinicians.

### 5.4. Concluded Prospective Studies

So far, as of December 2018, only two prospective, multicentre, open-label, randomized trials comparing the efficacy and safety of DOACs versus LMWHs in cancer patients with CAT have been fully published (Hokusai VTE Cancer, Select-D), and one has been presented in abstract form (ADAM VTE, Table 1).

#### 5.4.1. Edoxaban: The Hokusai VTE Cancer Clinical Trial

The Hokusai VTE cancer clinical trial, published in February 2018, compared a DOAC, edoxaban (at least 5 days of LMWHs followed by 60 mg daily), versus dalteparin (200 IU/kg daily for one month followed by a reduction at 150 IU/kg daily) [62]. Patients were treated for at least 6 months and up to 12 months.

In this non-inferiority trial, a total of 1050 patients, from 114 centers in 13 countries, were included from July 2015 to December 2016. With the exclusion of basal- or squamous-cell skin cancer, patients had to be diagnosed in the previous two years or had to present with an active cancer. Defined as having active cancer at diagnosis were patients with any treatment for malignancy in the previous six months, or a cancer not in complete remission (hematologic neoplasm), or with recurrent or metastatic disease. Patients were randomized to be treated with either edoxaban or dalteparin in a 1:1 ratio.

The primary outcome was a composite of recurrent VTE or major bleeding in the first 12 months after randomization. Major bleeding was defined according to the criteria of the International Society on Thrombosis and Haemostasis (ISTH) [63]. The primary outcome occurred in 12.8% of patients in the edoxaban group and in 13.5% of patients in the LMWHs group (HR 0.97, 95% CI 0.70–1.36), confirming the non-inferiority of edoxaban. Rate of recurrent VTE was 7.9% in the edoxaban arm and 11.3% in the dalteparin arm (HR 0.71, 95% CI 0.48–1.06), again without a statistically significant difference. In contrast, the rate of major bleeding was significant higher, 6.9%, for the edoxaban arm and 4.0% for the dalteparin arm (HR 1.77, 95% CI 1.03–3.04). However, the higher rate of bleeding was mainly due to an upper gastrointestinal origin who occurred in patients with gastrointestinal cancer at enrollment. The rate of severe major bleedings (considered as a clinical emergency) was the same in the two groups (2.3%). Intracranial bleeding occurred in two patients treated with edoxaban and in four patients treated with dalteparin. Two patients died of major bleeding in the dalteparin group. There were no fatal VTE in both arms. Even if the rate of CRNMB, defined according to the criteria of the ISTH [63], was higher under edoxaban (14.6%) in comparison to dalteparin (11.1%), this was not statistically significant. There were no differences in both groups in terms of overall survival. In summary, edoxaban was non-inferior to dalteparin in the composite outcome of recurrent venous thromboembolism or major bleeding.

#### 5.4.2. Rivaroxaban: The SELECT-D Trial

Data on the efficacy and safety on the use of rivaroxaban in cancer patients was published in July 2018 in the SELECT-D trial, a pilot trial conducted in the United Kingdom [64]. Recruited between September 2013 and December 2016, the study reported 406 patients with active cancer at inclusion who presented not only symptomatic or incidental PE or symptomatic lower-extremity proximal DVT, but all locations of thrombotic were considered. Patients were randomly assigned to receive either rivaroxaban (15 mg twice daily for 21 days followed by 20 mg daily) or dalteparin (200 IU/kg daily for the first 30 days, followed by 150 IU/kg) for a total of six months. Active cancer was defined in the same way as in the Hokusai VTE cancer study. Both arms were well balanced (58% of patients with metastatic disease in both arms) and all locations of thrombotic events were considered (not only DVT or PE).

The main objective of this study was to assess VTE recurrence and safety (rates of major bleedings and CRNMB) in the first six months of treatment. The second objective of the study, which was to assess the treatment duration beyond six months, remained unanswered because of low recruitment at the second planned randomization (lower assignment due to high mortality rate and clinician’s decision).

The cumulative VTE recurrence rate was 11% in patients receiving dalteparin and 4% in patients receiving rivaroxaban (HR 0.43, 95% CI 0.19–0.99) at six months. Concerning safety, the cumulative major bleeds rate at six months was 6% in the rivaroxaban arm and 4% in the dalteparin arm (HR 1.83, 95% CI 0.68–4.96). There were no central nervous system (CNS) bleeds and most of major bleeding was of gastro-intestinal origin. In particular, patients with esophageal or gastroesophageal cancer had more bleeds under rivaroxaban than with dalteparin (four of 11 (36%) versus one of 19 (11%)). The cumulative rate of CRNMB at six months was 13% for rivaroxaban compared to 4% for dalteparin (HR 3.76, 95% CI 1.63–8.69). Most of CRNMB had a gastrointestinal or urologic origin. One fatal PE and one fatal major bleed occurred in each arm. No statistically significant differences of overall survival were noted in either the two arms at six months (70% for dalteparin and 75% for rivaroxaban). To conclude, the recurrence rate of VTE was lower under rivaroxaban, at the cost, however, of an increased rate of CRNMB.

#### 5.4.3. Meta-Analysis of the Prospective Studies

A meta-analysis of these two prospective trials confirmed the conclusions of both studies. At 6 months, patients with cancer treated with DOACs had a lower recurrence rate of VTE (42/725) compared to patients treated with LMWHs (64/727) (risk ratio [RR] 0.65, 95% CI 0.42–1.01). On the other hand, patients under DOACs presented a higher rate of major bleeding (40/725) in comparison to patients under LMWHs (23/727) (RR 1.74, 95% CI 1.05–2.88). Moreover, patients treated with DOACs showed a higher CRNMB rate (RR 2.31, 95% CI 0.85–6.28) [61]. At least three systematic reviews and network meta-analysis including all clinical trials, namely comprising the two prospective studies, were published in 2018 [65,66,67]. According to these results DOACs were superior in decreasing VTE recurrence, but an increased risk of major bleeding or CRNMB could not be excluded.

### 5.5. Ongoing or in 2018 Concluded Clinical Trials

Table 1 summarizes ongoing trials and two in 2018 completed studies, on the utilization of DOACSs in cancer patients diagnosed with VTE, based on a search of the Website https://clinicaltrials.gov [68]. Most of the studies are conducted with rivaroxaban, only two with apixaban and dabigatran respectively. Evaluating the role of rivaroxaban in the prevention and treatment of CAT is the goal of an international collaboration–the Cancer Associated thrombosis expLoring soLutions for patients through Treatment and prevention with rivaroxaban, abbreviated CALLISTO program. The studies CASTA-DIVA and COSIMO belong to this program. A trial that will possibly reflect the real-life situation, comprises all four DOACs in comparison to LMWHs.

## 6. Anticancer Effect

### 6.1. Anticancer Effect of Heparins

A potential anti-tumoral activity of unfractionated heparin has been suspected for more than 60 years [69]. The overall survival was beneficially influenced by UFH according to the data, published in 1998, obtained from some retrospective analyses of cancer patients included in randomized clinical trials comparing thromboprophylaxis with UFH versus no prophylaxis [70]. Two studies published one year later gave contradictory results. The first one, published by Lebeau and colleagues, found a statistically significant amelioration of survival in patients diagnosed with small cell carcinoma of the lung (SCLC) who received UFH (therapeutically dosed) in combination with chemotherapy versus without UFH in a prospective randomized multicentre trial [71]. The second one, a systematic review of clinical studies edited by Smorenburg and colleagues, found no solid evidence of a survival advantage in patients with cancer without VTE who were treated with UFH versus no treatment or placebo [72]. The same year, a meta-analysis evaluating the efficacy of LMWHs versus UFH in cancer patients with VTE, showed a lower mortality in patients initially treated with LMWHs [73]. Yet seven years earlier, in 1992, Prandoni and colleagues [74], as well as Green and colleagues [17], provided evidence of a survival advantage of LMWHs compared to UFH in cancer A potential survival advantage of anticoagulation in cancer patients was more recently reviewed [75,76,77]. The positive effect on survival relies not only on the anticoagulation action but also on a putative direct antineoplastic effect, in particular of LMWHs. For instance, a post-hoc analysis of the CLOT trial showed a mortality decrease at twelve months in patients without metastatic disease treated with dalteparin (20% versus 36%, HR 0.50, 95% CI 0.27–0.95) [78].

Many data, frequently very interesting and convincing, obtained from experimental models demonstrated a potential anticancer effect of heparins through multiple mechanisms [79,80,81]. Several biological mechanisms, which could explain the direct antitumoral activity of LMWHs were elucidated. They comprise: 1. inhibition of angiogenesis (by interfering with TF-FVIIa activation and with thrombin), 2. inhibition of P- and L-selectin-mediated cell adhesion, 3. inhibition of tumour invasion (by lowering heparanase activity), 4. inhibition of thrombin and fibrin production, 5. modulation of the immune system, 6. interference with tumor cell glycosaminoglycans, 7. induction of apoptosis, 8. some evidence of potential interference with cancer cell proliferation [82,83,84,85].

From a clinical point of view, alongside studies showing the efficacy of LMWHs for the treatment of CAT [23], analyses evaluating the effect of LMWHs in a cancer population without VTE have also been conducted [86,87]. The first two studies which showed a statistically significant improvement of overall survival in cancer patients treated with LMWHs were published in 2004 by Altinbas and colleagues [87], and by Kakkar and colleagues (FAMOUS study) [86]. Even if in the latter study the survival advantage was not statistically significant, a post hoc analysis in the subgroups of patients with a better prognosis at enrolment who received LMWHs had a statistically significant improvement in survival. In the following years at least nine other studies appeared showing conflicting results (i.e., MALT, PROTECHT, SAVE-ONCO, PRODIGE, TOPIC-I and -II, INPACT and others) [88,89,90,91,92,93,94,95,96]. Finally, the results of two randomised multicentre phase III controlled trials addressing the question of a survival advantage of LMWHs on a single cancer type—lung—were published in 2016 and 2018. The first one, the FRAGMATIC trial, did not show an improvement in overall survival [97]. The very recently published (October 2018) study, the TILT trial was not able to demonstrate a beneficial effect of overall survival or cancer recurrence even in early-stage lung cancer [98]. All in all, while the results of some clinical trials and meta-analysis recognized a statistically significant effect of LMWHs on survival [75,76,86,87,88,96], others found opposite conclusions [89,90,91,92,93,94,95,99,100]. Many reasons can explain this divergence. Some of them are the heterogeneity of studies design, of type and stage of included cancers, of used chemotherapy regimens and the type, dosing and duration of the treatment of LMWHs.

As a conclusion, despite the clinical evidence and the available experimental data, the two most recent meta-analyses did not observe a straightforward advantage of LMWHs. The first one, a 2014 meta-analysis of nine randomized trials, did not show a survival benefit of anticoagulation with LMWHs (odds ratio (OR) for one-year mortality 0.87, 95% CI 0.70–1.08) [99]. The second one, the Cochrane review published the same year, including 15 randomized trials comparing heparin with no intervention or placebo, showed a borderline effect on survival, a decrease in VTE incidence and a rise in minor bleeding [101]. Accordingly, the use of anticoagulation to exert an antineoplastic effect is not supported by evidence coming from these meta-analyses and Cochrane reviews. According to different international guidelines, anticoagulation in absence of VTE is not an option in the attempt to improve survival in cancer patients.

### 6.2. DOACs Beyond the Anticoagulation: A Potential Antineoplastic Effect?

In analogy to research looking for a potential antineoplastic role of LMWHs [80,102], some investigations indicate a potential effect of DOACs in this setting as well. The effects of DOACs beyond their principal role of inhibiting specific steps in the coagulation cascade eventually determining the anticoagulation state were summarized in 2014 [103].

Among many other mechanisms, cancer relies on angiogenesis to develop. Blocking this process, malignancy development could be limited. Angiogenesis is one of the effects of the tight interplay between inflammation and hemostasis [104,105], whose molecular link is constituted by protease-activated receptors (PARs).

Inflammation induces a procoagulant state in different ways (e.g. by reducing natural anticoagulants, by activating platelets, by releasing microparticles, by elevating the expression of the tissue factor (TF), and through NETosis) [106,107]. Conversely, the inflammation process is influenced by the coagulation cascade. PARs are transmembrane G-protein-coupled receptors that were discovered at the beginning of the nineties of the last century [108,109,110]. Fours PARs (PAR 1–4) have been identified and are expressed on cancer cells, on the cell surface of hematopoietic cells (leucocytes and platelets, but not on erythrocytes) and on several other cells (endothelial cells, vascular smooth muscle cells, fibroblasts and dendritic cells). Platelets express only PAR-1 and PAR-4, the first receptor having high affinity and the second having low affinity to the activator. The activation of the PAR-2 receptors induces prolonged secondary signaling. Thrombin acts as a protease on the extracellular domain of the N-terminus of the receptor. The so formed new N-terminal domain binds to a specific region of the extracellular part of the receptor inducing transmembrane signaling through G-proteins leading eventually to cell proliferation, activation and adhesion [109].

Thrombin influence many steps of cancer progression and seems to be a key player in this setting [111,112]. Coagulation protease, such as factor Xa and thrombin, by cleaving PARs on cell membranes trigger intracellular signaling events [113,114]. Thrombin can activate all types of PARs with the exception of PAR-2. Factor Xa, alone or bound to the complex TF-Factor VIIa, can activate PAR-1 and PAR-2. Thrombin and Factor Xa, through the interaction with PARs, eventually favor angiogenesis, inflammation and tissue fibrosis, thus stimulating tumor progression (Figure 1) [115]. Wojtukiewicz and colleagues summarized some of the scientific evidence pointing out the key role of thrombin in the tumor progression in 2017 [116].

Data on animal models suggest an anti-angiogenic effect of rivaroxaban at least as powerful as LMWHs [117]. Dabigatran is able to block free as well clot-bound thrombin. The latter favors angiogenesis and tissue repair at sites of injury [118]. In similar way, evidence from animal models or studies on cell lines on the inhibition of thrombin showed a diminution of tumor progression [119,120,121,122,123].

Thus, by inhibiting factor Xa and thrombin, it is theoretically conceivable to exert other effects than merely the anticoagulating one. By interfering with these processes, DOACs could exert an anti-neoplastic action. Whether this theoretic effect can traduce in clinical practice, needs to be investigated. Studies evaluating the risk/benefit ratio of DOACs for primary prevention VTE in intermediate-to-high-risk cancer patients, such as the recently published AVERT trial [124], could possibly contribute to this knowledge.

## 7. Conclusions

The recommended treatment for cancer-associated venous thromboembolism is based on LMWHs. Increasing clinical evidence places DOACs in a position to represent a real treatment option for CAT. However, their possible benefits have to be carefully balanced with the disadvantages of DOACs in the special setting of CAT.

The main advantages include an ease way of administration, a reduction in treatment costs, an absence of monitoring with the effect of reducing VTE recurrence. Inconveniences with the treatment of DOACs consist of an increased risk of bleeding (e.g., patients with gastro-intestinal or urologic malignancy, chemotherapy-induced thrombocytopenia) and a considerable interaction with other drugs. The latter is particularly sensitive to a rapidly changing landscape of cancer treatment. Reduced renal function and extreme body weights are limiting factors for the use of both LMWHs and DOACs. In these special circumstances, monitoring plasma trough-concentrations of the chosen anticoagulant is a prudent way to warrant safety. The preferences and the values of the patient are also to be included in the decision-making process of defining the type of anticoagulation.

Considering the different mechanism of actions (inhibition of thrombin or factor Xa) and pharmacokinetics aspects (clearance through P-glycoprotein and/or cytochrome P450), a uniformly effect of the class of DOACs cannot be assumed. Randomized clinical trials evaluating efficacy and safety for each DOACs versus LMWHs are therefore indispensable. Results of such studies are eagerly awaited in the next few years. A direct anti-neoplastic effect of DOACs is an interesting field of research that needs to be deepened before drawing conclusions on clinical efficacy.

In summary, the treatment of CAT must be personalized and adapted to the specific clinical condition of the patient. Based on the currently available data for edoxaban and rivaroxaban, DOACs represent an additional therapeutic option for patients with CAT and a low risk of bleeding.

## Figures and Tables

**Figure 1 cancers-11-00046-f001:**
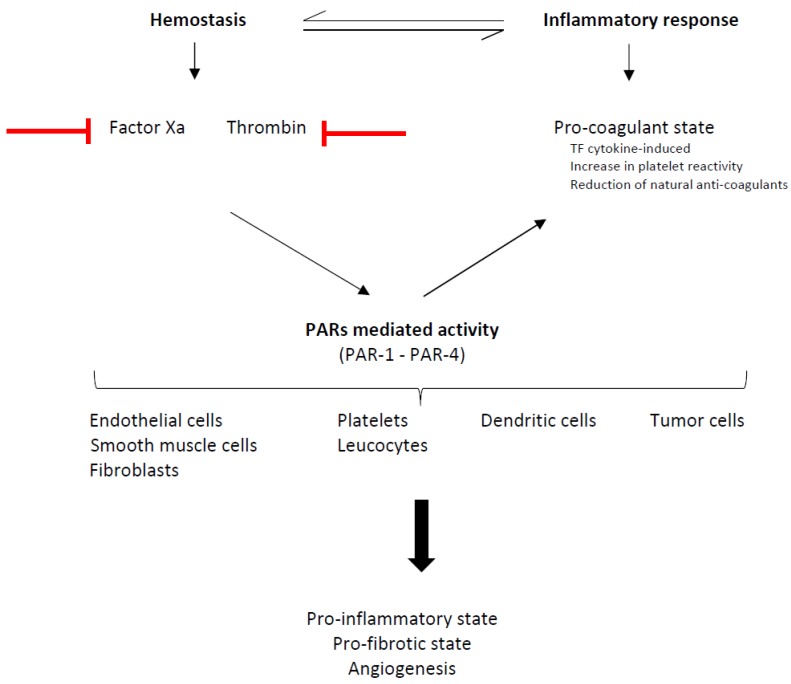
Schematic representation of the interaction between hemostasis and inflammation promoting tumor progression. PARs (protease-activated receptors) are expressed on several cells. Thrombin activates PAR-1, PAR-3 and PAR-4. Factor Xa, alone or bound to the tissue factor (TF)–activated factor VII (VIIa) complex, activates PAR-1 and PAR-2. In red: inhibition of either thrombin or factor Xa could potentially inhibit cancer progression.

**Table 1 cancers-11-00046-t001:** Summary of the ongoing or in 2018 concluded clinical trials.

DOACs vs. LMWHs	Diagnosis for Inclusion	Design, Phase, Estimated Number of Included Patients	Name of the Study	NCT, State, Estimated Completion Date
**Rivaroxaban**				
vs. low molecular weight heparins (LMWHs)	Venous Thromboembolism (VTE)	RCT phase III 450	CONKO_011/AIO-SUP-0115/Ass.: Rivaroxaban in the Treatment of Venous Thromboembolism (VTE) in Cancer Patients—a Randomized Phase III Study	02583191 Recruiting December 2018
Rivaroxaban only	Pulmonary embolism (PE) and Deep Vein Thrombosis (DVT)	Prospective phase III 500	A Non-interventional Study on Xarelto for Treatment of Venous Thromboembolism (VTE) and Prevention of Recurrent VTE in Patients with Active Cancer (COSIMO)	02742623 Active, not recruiting 15.03.2019
Rivaroxaban only	PE and DVT of upper and lower extremities	Retrospective NA 375	Rivaroxaban Utilization for Treatment and Prevention of Thromboembolism in Cancer Patients: Experience at a Comprehensive Cancer Center	02502396 Active, not recruiting September 2020
vs. LMWHs	VTE	RCT 200	Cancer Associated Thrombosis, a Pilot Treatment Study Using Rivaroxaban (CASTA-DIVA)	02746185 Completed 25.04.2018
vs. Dalteparin	VTE	RCT phase II 176	A Randomized Phase II Study to Compare the Safety and Efficacy of Dalteparin vs. Rivaroxaban for Cancer-associated Venous Thromboembolism (PRIORITY)	03139487 Recruiting May 2020
**Apixaban**				
vs. LMWHs	VTE	RCT phase III 1168	Apixaban for the Treatment of Venous Thromboembolism in Patients with Cancer (CARAVAGGIO)	03045406 Recruiting June 2019
vs. Dalteparin	VTE	RCT Phase III 315	Apixaban or Dalteparin in Reducing Blood Clots in Patients with Cancer Related Venous Thromboembolism (ADAM VTE)	02585713 Completed November 2018
Apixaban only	Venous Thrombosis	Single Group Assignment 300	Apixaban as Treatment of Venous Thrombosis in Patients with Cancer: The CAP Study (CAP)	02581176 Completed May 2018
**Dabigatran**				
vs. tinzaparin	VTE	Phase III 99	A Study of Dabigatran Etexilate as Primary Treatment of Malignancy Associated Venous Thromboembolism	03240120 Recruiting 31.12.2021
**All DOACS (apixaban, edoxaban, dabigatran, rivaroxaban)**				
vs. LMWHS alone or with warfarin	VTE (cumulative recurrence)	RCT NA 940	Direct oral anticoagulants (DOACs) vs. LMWH+/−warfarin for VTE in cancer: a randomized effectiveness trial (CANVAS TriaL)	02744092 Recruiting September 2019

NCT: Number of Clinical Trials. RCT: Randomized Clinical Trial. NA: Not Applicable.

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
