# Peer review of "Direct Oral Anticoagulant Drugs: On the Treatment of Cancer-Related Venous Thromboembolism and their Potential Anti-Neoplastic Effect"

_cancers, 2019, doi:10.3390/cancers11010046_

Round 1
Reviewer 1 Report
Overall, this is a timely and balanced review article on the role of DOACs in the treatment of cancer-associated VTE. In addition to reiterating important clinical trial data, the article also touches on potential antineoplastic effects of the direct oral FXa and thrombin inhibitors, which is an innovative and somewhat provocative concept.
I have the following comments:
From a logical point of view, I would first comment on clinical trial data and then speculate on the anticancer effects of DOACs. Thus, the authors may consider changing the order of their manuscript’s chapters.
A thromboembolic event is not identical to VTE, since the former term also includes arterial thromboembolism (e.g. stroke). The abbreviation VTE should only be used for venous thromboembolism; thromboembolic events should be abbreviated as TEs. In addition, the abbreviation CAT is commonly used for cancer-associated thrombosis (or cancer-associated thromboembolism). Please check your manuscript for consistency.
Page 2, lines 76-78: It should be "… use of edoxaban or rivaroxaban …" to avoid any misunderstanding (some readers might think that both agents should be used together). Furthermore, "DOACs should only be administered …" (instead of "could").
Page 2, last paragraph: I don’t think that the study by Prandoni et al. showed a six-fold increased risk of bleeding in cancer patients compared to non-cancer patients. To the best of my knowledge, rates of major bleeding were 12.4 % and 4.9 % in cancer and non-cancer patients, respectively.
Some sentences appear incomplete. For instance, page 3, first paragraph ("The first randomized clinical study …") or page 8, fifth paragraph, lines 328-332.
Regarding the anticancer effects of LMWH (page 3), specific earlier (e.g. FAMOUS, MALT, INPACT) and more recent trials investigating the effect of LMWH on the survival of cancer patients should be cited in addition to review articles and meta-analyses.
Page 4, first paragraph: Add "stroke" to "systemic embolization" (line 146). I think that the wording "must not be treated" is too rigorous for patients with extensive DVT or severe renal impairment (which does not necessarily mean that the patient requires dialysis).
Page 4, fourth paragraph (line 159): What does "slightly superior" mean? Apixaban significantly reduced both major and clinically relevant (i.e. major and clinically relevant non-major) bleeding compared to warfarin.
Page 4, sixth paragraph (line 165): RECOVER II did not only include patients with acute DVT.
Page 4, second-to-last paragraph (line 190): Dabigatran is not administered as a tablet. Later, the authors correctly point out that dabigatran is administered as a capsule.
Page 5: In my opinion, it is not common to use the terms PARs 1 and PARs 2. I suggest only using PAR-1 and PAR-2. Also see Fig. 1. This figure appears to be a screen shot from a file opened in correction mode.
Page 6, first paragraph (lines 238 and 239): Rivaroxaban does not inhibit clot-bound thrombin.
Page 6, line 247: Change DVT to VTE.
Please carefully check your numbers for accuracy. For example, in the pooled analysis of the EINSTEIN trials involving patients with cancer diagnosed during the study, rates of recurrent VTE were 10 % and 12 % in patients receiving rivaroxaban and VKA, respectively (and not 20 % and 12 %).
Page 9, second paragraph: I wouldn’t say that SELECT-D was a non-inferiority trial. The authors later correctly point out that SELECT-D was an exploratory pilot trial. HOKUSAI VTE Cancer and SELECT-D should not be "put in the same pot".
Page 9, third paragraph (line 368): Change "5 days" to "at least 5 days". Some patients in HOKUSAI VTE Cancer received LMWH for much longer than 5 days before being switched to edoxaban (and this was allowed by the study protocol).
Page 10, first paragraph: Please specify that "all locations of thrombotic events were considered" refers to outcome and not to index VTE events (at least to the best of my knowledge).
Page 10, fourth paragraph (line 422): The VTE recurrence rate was not inferior under rivaroxaban, but lower.
Page 12, third paragraph (line 461): Clearance through P-GP and/or CYP3A4 are pharmacokinetic (and not pharmacodynamic dynamic) aspects.
I am missing the ADAM VTE trial investigating apixaban in comparison to dalteparin in patients with cancer-associated VTE. This study should either be added to Table 1 or discussed in the text.
Author Response
CAT and DOACs_Cancers-408832
Reviewer 1
Overall, this is a timely and balanced review article on the role of DOACs in the treatment of cancer-associated VTE. In addition to reiterating important clinical trial data, the article also touches on potential antineoplastic effects of the direct oral FXa and thrombin inhibitors, which is an innovative and somewhat provocative concept.
Response:
We appreciated the comments of the reviewer who helped us to improve the quality of the manuscript. Thank you for the constructive criticism!
I have the following comments:
1.1. From a logical point of view, I would first comment on clinical trial data and then speculate on the anticancer effects of DOACs. Thus, the authors may consider changing the order of their manuscript’s chapters.
Response:
We agree with the recommendation and changed the order of the chapters.
1.2. A thromboembolic event is not identical to VTE, since the former term also includes arterial thromboembolism (e.g. stroke). The abbreviation VTE should only be used for venous thromboembolism; thromboembolic events should be abbreviated as TEs. In addition, the abbreviation CAT is commonly used for cancer-associated thrombosis (or cancer-associated thromboembolism). Please check your manuscript for consistency.
Response:
All abbreviations were checked as requested.
1.3. Page 2, lines 76-78: It should be "… use of edoxaban or rivaroxaban …" to avoid any misunderstanding (some readers might think that both agents should be used together). Furthermore, "DOACs should only be administered …" (instead of "could").
Response:
Corrected as requested.
1.4. age 2, last paragraph: I don’t think that the study by Prandoni et al. showed a six-fold increased risk of bleeding in cancer patients compared to non-cancer patients. To the best of my knowledge, rates of major bleeding were 12.4 % and 4.9 % in cancer and non-cancer patients, respectively.
Response: Thank you for identifying the mistake; corrected as requested based on the data of Prandoni (HR of 3.2 for thromboembolism recurrence, and of 2.2 for major-bleeding).
1.5. Some sentences appear incomplete. For instance, page 3, first paragraph ("The first randomized clinical study …") or page 8, fifth paragraph, lines 328-332.
Response:
Corrected as requested. Page 3, first paragraph, simplified. Page 8, fifth paragraph: removed as too detailed.
1.6. Regarding the anticancer effects of LMWH (page 3), specific earlier (e.g. FAMOUS, MALT, INPACT) and more recent trials investigating the effect of LMWH on the survival of cancer patients should be cited in addition to review articles and meta-analyses.
Response:
This paragraph was revised in depth adding more clinical data. A short paragraph on the biological mechanism of LMWHs was added, as well.
1.7. Page 4, first paragraph: Add "stroke" to "systemic embolization" (line 146). I think that the wording "must not be treated" is too rigorous for patients with extensive DVT or severe renal impairment (which does not necessarily mean that the patient requires dialysis).
Response:
“Stroke” added as requested. “Must not” replaced by “should not”.
1.8. Page 4, fourth paragraph (line 159): What does "slightly superior" mean? Apixaban significantly reduced both major and clinically relevant (i.e. major and clinically relevant non-major) bleeding compared to warfarin.
Response:
“SlightlY” cancelled.
1.9. Page 4, sixth paragraph (line 165): RECOVER II did not only include patients with acute DVT.
Response:
DVT replaced by VTE as requested.
1.10. Page 4, second-to-last paragraph (line 190): Dabigatran is not administered as a tablet. Later, the authors correctly point out that dabigatran is administered as a capsule.
Response:
Corrected as requested.
1.11. Page 5: In my opinion, it is not common to use the terms PARs 1 and PARs 2. I suggest only using PAR-1 and PAR-2. Also see Fig. 1. This figure appears to be a screen shot from a file opened in correction mode.
Response: Corrected as requested.
1.12. Page 6, first paragraph (lines 238 and 239): Rivaroxaban does not inhibit clot-bound thrombin.
Response: Corrected as requested.
1.13. Page 6, line 247: Change DVT to VTE.
Response:
Corrected as requested.
1.14. Please carefully check your numbers for accuracy. For example, in the pooled analysis of the EINSTEIN trials involving patients with cancer diagnosed during the study, rates of recurrent VTE were 10 % and 12 % in patients receiving rivaroxaban and VKA, respectively (and not 20 % and 12 %).
Response: We checked numbers through all the manuscript.
1.15. Page 9, second paragraph: I wouldn’t say that SELECT-D was a non-inferiority trial. The authors later correctly point out that SELECT-D was an exploratory pilot trial. HOKUSAI VTE Cancer and SELECT-D should not be "put in the same pot".
Response: We agree; corrected being only the HOKUSAI a non-inferiority one.
1.16. Page 9, third paragraph (line 368): Change "5 days" to "at least 5 days". Some patients in HOKUSAI VTE Cancer received LMWH for much longer than 5 days before being switched to edoxaban (and this was allowed by the study protocol).
Response: Corrected as requested.
1.17. Page 10, first paragraph: Please specify that "all locations of thrombotic events were considered" refers to outcome and not to index VTE events (at least to the best of my knowledge).
Response:
Changed as follows: “[…] study reported 406 patients with active cancer at inclusion who presented not only symptomatic or incidental PE or symptomatic lower-extremity proximal DVT, but all locations of thrombotic were considered.”
1.18. Page 10, fourth paragraph (line 422): The VTE recurrence rate was not inferior under rivaroxaban, but lower.
Response:
Corrected.
1.19. Page 12, third paragraph (line 461): Clearance through P-GP and/or CYP3A4 are pharmacokinetic (and not pharmacodynamic dynamic) aspects.
Response:
Yes, indeed; corrected.
1.20. I am missing the ADAM VTE trial investigating apixaban in comparison to dalteparin in patients with cancer-associated VTE. This study should either be added to Table 1 or discussed in the text.
Response:
Added to Table 1.
Reviewer 2 Report
The effects of direct oral anticoagulant drugs (DOACs) on the treatment of cancer-related venous thromboembolism and their potential anti-neoplastic effect were reviewed in this paper. Is there any evidence showing that DOACs have any differential anticoagulant effects or anti-neoplastic effects on different solid malignant tumors, leukemia, and malignant lymphoma? This should be briefly introduced in this manuscript.
Author Response
CAT and DOACs_Cancers-408832
Reviewer 2
The effects of direct oral anticoagulant drugs (DOACs) on the treatment of cancer-related venous thromboembolism and their potential anti-neoplastic effect were reviewed in this paper. Is there any evidence showing that DOACs have any differential anticoagulant effects or anti-neoplastic effects on different solid malignant tumors, leukemia, and malignant lymphoma? This should be briefly introduced in this manuscript.
Response:
No evidence showing that DOACs have a differential anticoagulant or anti-tumor efficacy on different neoplasms is published so far, to the best of our knowledge. However, the results of the SELECT-D trials showed that the site of primary tumor (stomach or pancreas versus other) predicted for VTE recurrence and VTE type [Young et al. J Clin Oncol. 2018;36(20):2017-23]. So far, no studies addressing the question of a differential anticoagulant or anti-tumor effect were conducted with DOACs. Should the reviewer be aware of any work addressing this topic, we will be pleased to insert it in our manuscript.
Reviewer 3 Report
In this manuscript, Dr. Grandoni and Dr. Alberio nicely reviewed the literature about the use of DOACs in cancer patients. In addition to the anti-thromboembolism effect of DOACs, this manuscript also discussed the and anti-cancer effect of DOACs. Overall, this review is very detailed and comprehensive.
I have one major concern. The authors point out the disadvantage of heparin treatment, including high cost, painful injections, poor long-term outcome, and HIT. To summarize the disadvantage or any published side effects of DOACs may be also helpful.
Author Response
CAT and DOACs_Cancers-408832
Reviewer 03
In this manuscript, Dr. Grandoni and Dr. Alberio nicely reviewed the literature about the use of DOACs in cancer patients. In addition to the anti-thromboembolism effect of DOACs, this manuscript also discussed the anti-cancer effect of DOACs. Overall, this review is very detailed and comprehensive.
I have one major concern. The authors point out the disadvantage of heparin treatment, including high cost, painful injections, poor long-term outcome, and HIT. To summarize the disadvantage or any published side effects of DOACs may be also helpful.
Response:
We thank the reviewer for his/her appreciation of our manuscript.
The adverse effects and concerns with the use of DOACs were summarized in Chapter 4.
Round 2
Reviewer 1 Report
The paper is now acceptable for publication!
Reviewer 3 Report
I have no further questions.